# Prolonged Length of Stay in the Emergency Department and Increased Risk of In-Hospital Cardiac Arrest: A nationwide Population-Based Study in South Korea, 2016–2017

**DOI:** 10.3390/jcm9072284

**Published:** 2020-07-18

**Authors:** June-sung Kim, Dong Woo Seo, Youn-Jung Kim, Jinwoo Jeong, Hyunggoo Kang, Kap Su Han, Su Jin Kim, Sung Woo Lee, Shin Ahn, Won Young Kim

**Affiliations:** 1Department of Emergency Medicine, University of Ulsan, College of Medicine, Asan Medical Center, Seoul 05505, Korea; jsmeet09@gmail.com (J.-s.K.); leiseo@gmail.com (D.W.S.); yjkim.em@gmail.com (Y.-J.K.); ans1023@gmail.com (S.A.); 2Department of Biomedical Informatics, University of Ulsan, College of Medicine, Asan Medical Center, Seoul 05505, Korea; 3Department of Emergency Medicine, Dong-A University, College of Medicine, Busan 61656, Korea; advanced@lifesupport.pe.kr; 4Department of Emergency Medicine, Hanyang University, College of Medicine, Seoul 04763, Korea; emer0905@gmail.com; 5Department of Emergency Medicine, Korea University, College of Medicine, Seoul 02841, Korea; hanks96@hanmail.net (K.S.H.); icarusksj@gmail.com (S.J.K.); kuedlee@korea.ac.kr (S.W.L.)

**Keywords:** emergency department crowding, in-hospital cardiac arrest, quality control, length of stay

## Abstract

This study was to determine whether prolonged emergency department (ED) length of stay (LOS) is associated with increased risk of in-hospital cardiac arrest (IHCA). A retrospective cohort with a nationwide database of all adult patients who visited the EDs in South Korea between January 2016 and December 2017 was performed. A total of 18,217,034 patients visited an ED during the study period. The median ED LOS was 2.5 h. IHCA occurred in 9,180 patients (0.2%). IHCA was associated with longer ED LOS (4.2 vs. 2.5 h), and higher rates of intensive care unit (ICU) admission (58.6% vs. 4.7%) and in-hospital mortality (35.7% vs. 1.5%). The ED LOS correlated positively with the development of IHCA (Spearman ρ = 0.91; *p* < 0.01) and was an independent risk factor for IHCA (odds ratio (OR) 1.10; 95% confidence interval (CI), 1.10–1.10). The development of IHCA increased in a stepwise fashion across increasing quartiles of ED LOS, with ORs for the second, third, and fourth relative to the first being 3.35 (95% CI, 3.26–3.44), 3.974 (95% CI, 3.89–4.06), and 4.97 (95% CI, 4.89–5.05), respectively. ED LOS should be reduced to prevent adverse events in patients visiting the ED.

## 1. Introduction

In-hospital cardiac arrest (IHCA) is an acute episode that can occur in hospitalized patients during an emergency department (ED) stay or after admission [1]. The incidence of IHCA in adults has been reported to range from six to eight cases per 1,000 admissions, although the true incidence remains unclear [2,3]. Many IHCAs are unexpected, but others can be avoided by early identification and adequate management of at-risk patients [4]. Caring for critically ill patients is challenging, and requires delicate monitoring and interventions. Because ED care focuses more on prompt diagnosis and stabilization than on detail care of the patient, as in the intensive care unit (ICU) or admission as an inpatient, a more extended stay in the ED could increase the risk of adverse events [5]. Moreover, emergency physicians and nurses are rarely educated in care of critically ill patients, and patient-to-nurse ratios are quite higher in the ED than in the intensive care unit (ICU) which may lead to poor medical services [5].

Longer stay in the ED is a global trend, due largely to a mismatch between the supply of medical resources and increased demands on EDs [6]. Prolonged ED length of stay (LOS) has been associated with poor outcomes in various situations, including the development of dementia in older people and adverse cardiovascular outcomes in patients with acute coronary syndrome [7,8]. Moreover, the negative impact of prolonged ED LOS could be severe in critically ill patients with acute respiratory failure caused by pneumonia [9]. Although studies have assessed the relationship between prolonged ED stay and mortality [10], few large population-based studies have evaluated the direct association between increased ED LOS and prevalence of IHCA.

Based on the hypothesis that prolonged stay in the ED would increase the risk of sudden cardiac arrest, the current study was designed to evaluate the association between ED LOS and the development of IHCA.

## 2. Materials and Methods

### 2.1. Study Design and Setting

This nationwide population-based observational study used data from the National Emergency Department Information System (NEDIS) of South Korea. NEDIS was launched in 2003 by the Ministry of Health and Welfare, and was managed by the National Emergency Medical Center of Korea [11]. NEDIS covers all clinical and administrative data of patients who visited EDs throughout the country. Because the government annually monitors the quality of all data compiled by NEDIS and provides feedback to hospitals, NEDIS data should reflect data of all EDs in South Korea. This study did not require approval by the study facility’s ethics committee, and informed consent was waived because of the anonymity of NEDIS data.

### 2.2. Selection of Participants

This study included data on all adults, aged ≥ 18 years, who visited any ED in Korea between January 1, 2016 and December 31, 2017. Cardiac arrest was identified using ICD 10 codes corresponding to cardiac arrest or resuscitation, such as I46.0 (cardiac arrest with successful resuscitation), I46.1 (sudden cardiac death), and I46.9 (cardiac arrest, unspecified). Previous reports proved that these combinations of ICD codes could find CA with high specificity [12]. Patients who experienced out-of-hospital cardiac arrest were excluded; these included patients with ICD codes for cardiac arrest or resuscitation but without initially recorded vital signs, patients who arrived at the ED with unstable initial vital signs indicating peri-arrest status (systolic blood pressure ≤ 40 mmHg; heart rate ≤ 20 /min; and respiratory rate ≤ 4 /min), and patients who had cardiac arrest within 60 min of ED admission. In addition, we excluded patients who were under 18 years old; those with trauma; and those without any vital signs during ED stay.

### 2.3. Methods of Measurement

Demographic and clinical data included patient age, gender, initial vital signs on ED arrival, cancer, Korean Triage and Acuity Scale (KTAS), and ED LOS. Patients with cancer included those with both solid and hematologic malignancies, without staging information. The KTAS is a reliable tool for triaging patients according to severity [13]. Patients were assigned KTAS levels of 1 to 5, with level 1 indicating the most severely ill patients, including unstable vital signs and altered mentality. For convenience, severity of illness was dichotomized as KTAS level 1 or not, and mental status as Glasgow Coma Scale > 13 (alert) or not. ED LOS was defined as the interval between ED arrival and ED departure, with the latter including hospital admission, discharge from the ED, or transfer to another hospital. Other clinical outcomes included ICU admission and in-hospital mortality. Patients who experienced IHCA during ED stay and after admission were classified by ED LOS, and the incidence of cardiac arrest was calculated.

The primary outcome of the study was the association between ED LOS and the occurrence of IHCA during ED stay and after admission. The secondary outcome was the mortality rate according to the ED LOS. The date of the patient’s death was extracted from the National Health Insurance Service in South Korea.

### 2.4. Statistical Analyses

All statistical analyses were performed with R statistics software, version 3.5.0 (R Foundation for Statistical Computing, Vienna, Austria). The association between ED LOS and IHCA was evaluated by descriptive statistics. The normality of distribution was examined using the Kolmogorov–Smirnov test. Continuous variables are presented as medians with interquartile ranges (IQR), and compared by Student’s t-test or the Mann–Whitney U-test, as appropriate. Categorical variables are expressed as number (N) and percentage (%), and compared by chi-square tests. The relationship between ED LOS and IHCA was assessed by Spearman rank correlation coefficient, with the cut-off value of LOS determined by the Youden-index method. Classical methods were utilized to calculate the sensitivity, specificity, positive predictive value (PPV), and negative predictive value (NPV) of each LOS cut-off. A multivariate regression model that included potential confounders, such as age, gender, cancer, and KTAS, was used to determine whether ED LOS was an independent risk factor for IHCA. *P*-values < 0.05 were considered statistically significant.

## 3. Results

### 3.1. Baseline Characteristics of Study Population

During the study period, a total of 18,217,034 patients visited EDs throughout Korea; of these, 12,390,418 patients were excluded: 4,301,623 because they were aged < 18 years, 7,803,387 because they presented with trauma, 284,534 because initial vital signs were not recorded, and 874 because they experienced cardiac arrest within 60 min of arrival at the ED. Thus, 5,826,616 adult patients of median age 53.0 years (interquartile range (IQR), 37.0–68.0 years) and median ED LOS 2.5 h (IQR, 1.5–4.4 h) were analyzed. Of these 5,826,616 patients, 9,180 (0.2%) experienced IHCA: 5,946 (64.8%) during ED stay and 3,234 (35.2%) after hospital admission. The overall in-hospital mortality rate was 1.5%.

Table 1 shows the baseline characteristics of the study population. Patients who experienced IHCA were older (68.0 vs. 53.0 years) than those who did not. Male sex (65.8% vs. 46.6%) and cancer (11.2% vs. 7.0%) were more frequent in patients with than without IHCA. Mean arterial pressure at ED arrival was lower (87.3 vs. 96.3 mmHg) and the proportion of patients with KTAS level 1 (48.1% vs. 1.1%) was higher in patients who did than did not experience IHCA. Patients with IHCA had poorer clinical outcomes, including higher rates of ICU admission (58.6% vs. 4.7%) and in-hospital mortality (35.7% vs. 1.5%). ED LOS was longer in patients who did than did not experience IHCA (4.2 vs. 2.5 h). Similar results were observed when patients who did and did not experience IHCA in the ED, and patients who did and did not experience IHCA after admission were compared (Appendix A).

### 3.2. Main Results

Figure 1 shows a histogram comparing numbers of patients who experienced IHCA and ED LOS. Most cardiac arrests occurred within 2 h, markedly decreasing over time. However, a positive correlation between ED LOS and IHCA was observed when the incidence of IHCA was calculated by dividing the number of patients who experienced cardiac arrest by the total number of patients at each ED LOS interval (Spearman ρ = 0.91; *p* < 0.01; Figure 2). To determine the optimal ED LOS for predicting IHCA, ED LOS cutoffs of 3, 6, 9, and 12 h were tested (Table 2). An ED LOS cut-off of 3 h showed the highest sensitivity (65.0%), along with a specificity of 60.0%, a PPV of 0.0%, and an NPV of 100.0%. The 12-h cut-off showed the highest specificity (93.0%) but the lowest sensitivity (20.0%).

In the overall sample, several variables other than ED LOS, including older age, male gender, cancer comorbidity, and proportion of level 1 KTAS, were found to be associated with IHCA (Table 3). However, longer LOS remained an independent risk factor for IHCA (odds ratio (OR), 1.10; 95% confidence interval (CI), 1.10–1.10; *p* < 0.001). When these patients were divided into four LOS quartiles (<1.5, 1.5–2.5, 2.5–4.4, and > 4.4 h), longer ED LOS was associated with a stepwise increase in the risk of IHCA. Relative to the first (reference quartile), the ORs for the second, third, and fourth quartiles were 3.35 (95% CI, 3.26–3.44), 3.974 (95% CI, 3.89–4.06), and 4.97 (95% CI, 4.89–5.05), respectively (all *p* < 0.001).

ED LOS was significantly longer in patients who died in hospital than in those who survived (10.1 vs. 4.4 h; *p* < 0.01; Appendix A). Moreover, a multivariate logistic regression model that adjusted for age, gender, malignancy, and KTAS level 1 found that ED LOS was an independent predictor of survival in the total cohort (adjusted OR, 1.01; 95% CI, 1.01–1.01; *p* < 0.01; Appendix A).

## 4. Discussion

The current study found that longer ED LOS was correlated with the occurrence of IHCA during ED stay and after hospital admission. When adjusted for age, gender, malignancy, and severity, prolonged ED LOS was an independent risk factor for the development of cardiac arrest and showed a cumulative effect.

Increased ED LOS has been shown to correlate with delays in diagnosis and treatment, and with adverse patient outcomes [14,15]. Our results are in agreement with these findings, in that longer ED stay was associated with adverse outcomes. There are several possible reasons for this association. For example, the time required for laboratory and imaging results may be longer for more severely ill patients, and the occurrence of cardiac arrest may be due to more severe illness rather than a direct result of longer ED stay. Compared with patients who did not experience IHCA, those in our study population who experienced IHCA had higher proportions of older aged patients, patients with KTAS level 1 severity, and lower mean arterial pressure on arrival at the ED. These confounders, however, did not change our finding that prolonged stay in the ED was an independent risk factor for the increased occurrence of IHCA. In addition, time had a cumulative effect on IHCA.

Another reason may be due to ED crowding. Many studies have reported that ED crowding has unfavorable effects on patients. Although there have been no standards for measuring ED overcrowding, LOS is regarded as reflecting crowded EDs [16]. Overcrowding, resulting in shortages of space and human resources, can delay the adequate diagnosis and treatment [17]. Prolonged stay in the ED was also reported to be related to decreases in proper management of various diseases, including asthma, acute coronary syndrome, and pneumonia [18,19]. Moreover, a recent retrospective study revealed the correlation between ED crowding measured by bed occupancy rate and the incidence of IHCA [20]. They insisted that the proportion of bed occupancy could reflect the number of patients requiring bedrest in the EDs, and this index was associated with an increased risk of the occurrence of IHCA. Even though there was no evidence of direct correlation between ED occupancy rate and ED LOS, both measurements could imply similar negative clinical impacts on critically ill patients.

Our results also suggest that ED LOS is related to increased mortality. Even if there is no causal relationship between longer ED stay and mortality, longer ED stay has shown harmful consequences [21]. There were numerous studies reported about relationships between prolonged LOS and increased mortality in various situations [22,23]. Guttmann et al. analyzed a nationwide data of Canadian ED data and found that longer waiting patients in ED showed higher short-term mortality [22]. A more recent retrospective study also announced that ED LOS was an independent risk factor for in-hospital mortality among sepsis patients requiring ICU admission [23]. In addition, our finding suggested that higher mortality might be mainly due to a higher incidence of cardiac arrest. We found that the patients who experienced IHCA had higher ICU admission and mortality rates, a finding consistent with results showing poor outcomes in patients who experienced IHCA [24]. In addition, most EDs are designed to care for at-risk patients upon arrival, not for patients awaiting test results or admission.

One of the advantages of this study was the large sample size based on nationwide research. Moreover, to our knowledge, this is the first study showing a correlation not only between ED LOS and IHCA during ED stay, but also after hospital admission. Nevertheless, this study had several limitations, mostly stemming from its retrospective design. First, selecting patients who experienced cardiac arrest using ICD codes related to cardiac arrest and resuscitation may have omitted some cardiac arrest patients. We categorized the top 10 classifications of ED diagnosis according to the development of IHCA and found that primary cardiac and respiratory causes were dominant (Appendix A). Second, assuming peri-arrest vital signs on ED arrival may not be an accurate reflection of IHCA. However, the incidence of IHCA in the study population was about 0.2%, similar to previous studies. Third, we excluded the patients (874, 8.7% of total IHCA) who experienced IHCA within one hour because the primary purpose of our study was to reveal the association between prolonged LOS and adverse outcomes. We assumed that the development of IHCA within one hour was not influenced by a prolonged ED stay. Furthermore, most cardiac arrests that occurred within 60 min were initially unstable and could not be avoided although physicians tried to manage and resuscitate patients. A group of IHCA after 1 h had a lower proportion of Glasgow Coma Scale < 13, KTAS level 1, and ICU admissions than that of IHCA within 1 h, and these results supported our assumptions (Appendix A). Finally, although we tried to diminish confounders, such as severity of triage scale and malignancy, this study did not consider unmeasured confounders, including comorbidities, medications, and various quality of resuscitation across hospitals, which may have influenced the results. 

## 5. Conclusions

Increased ED LOS is an independent risk factor for the development of IHCA, both in the ED and after hospital admission. Our findings indicate the importance of reducing ED LOS in patients visiting the ED to prevent adverse events.

## Figures and Tables

**Figure 1 jcm-09-02284-f001:**
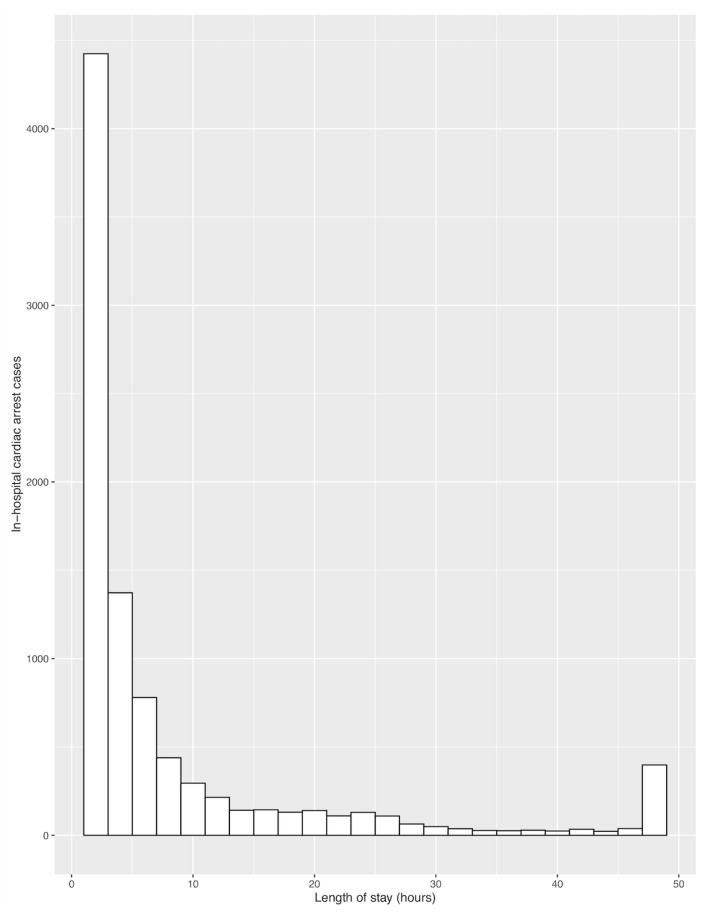
Relationship between in-hospital cardiac arrest and length of stay in the emergency department.

**Figure 2 jcm-09-02284-f002:**
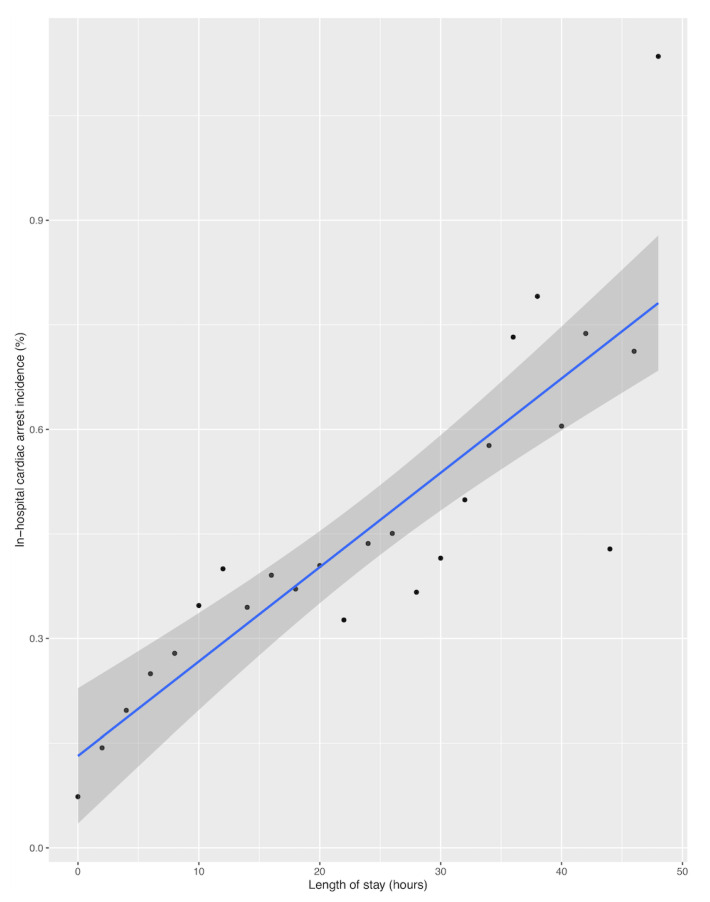
Correlation between length of stay in the emergency department and in-hospital cardiac arrest.

**Table 1 jcm-09-02284-t001:** Baseline demographic and clinical characteristics of emergency department patients with and without in-hospital cardiac arrest (IHCA).

Variables	Total (*N* = 5,826,616)	No IHCA (*N* = 5,817,436)	IHCA (*N* = 9,180)	*p*-Value
Age	53.0 (37.0–68.0)	53.0 (37.0–68.0)	68.0 (55.0–78.0)	< 0.01
Male	2,719,422 (46.7)	2,713,386 (46.6)	6036 (65.8)	< 0.01
MAP (mmHg) ^†^	96.3 (85.7–106.7)	96.3 (85.7–106.7)	87.3 (70.0–104.0)	< 0.01
Pulse rate, bpm ^†^	82.0 (74.0–96.0)	82.0 (74.0–96.0)	95.0 (78.0–114.0)	< 0.01
Respiratory rate, breaths per min ^†^	20.0 (18.0–20.0)	20.0 (18.0–20.0)	20.0 (18.0–22.0)	< 0.01
LOS in ED (hours)	2.5 (1.5–4.4)	2.5 (1.5–4.4)	4.2 (2.4–9.3)	< 0.01
Cancer	409,576 (7.0)	408,544 (7.0)	1,032 (11.2)	< 0.01
Not alert ^††^	231,663 (4.0)	226,657 (3.9)	5,006 (54.5)	< 0.01
KTAS level 1	69,838 (1.2)	65,422 (1.1)	4,416 (48.1)	< 0.01
ICU admission	278,001 (4.8)	272,626 (4.7)	5,375 (58.6)	< 0.01
Mortality	90,298 (1.5)	87,021 (1.5)	3,277 (35.7)	< 0.01

Data are presented as median (interquartile range (IQR)) or as number (percentage). ^†^ Vital signs were checked on ED arrival. ^††^ Glosgow Coma Scale < 13. Abbreviations: IHCA = in-hospital cardiac arrest; MAP = mean arterial pressure; ED = emergency department; KTAS = Korean Triage and Acuity Scale; ICU = intensive care unit.

**Table 2 jcm-09-02284-t002:** Sensitivity and specificity of each time cut-off of stay in the emergency department predicting in-hospital cardiac arrest (IHCA).

Cut-Off	Sensitivity	Specificity	PPV	NPV
3 h	0.65	0.60	0.0	1.0
6 h	0.37	0.84	0.0	1.0
9 h	0.26	0.91	0.0	1.0
12 h	0.20	0.93	0.0	1.0

Abbreviations: IHCA = in-hospital cardiac arrest; PPV = positive predictive value; NPV = negative predictive value.

**Table 3 jcm-09-02284-t003:** Multivariate analysis of factors associated with in-hospital cardiac arrest (IHCA).

Variable	OR	95% CI	*p*-Value
Age	1.02	1.01–1.02	< 0.01
Male	2.00	1.94–2.02	< 0.01
Cancer	1.26	1.15–1.29	< 0.01
KTAS level 1	61.04	60.99–61.08	< 0.01
ED LOS (hours)	1.10	1.10–1.10	< 0.01
Age	1.01	1.01–1.02	< 0.01
Male	1.92	1.88–1.97	< 0.01
Cancer	0.99	0.92–1.06	0.74
KTAS level 1	66.44	66.40–66.49	< 0.01
ED LOS quartile 1 (0–1.5 h)	Reference		
ED LOS quartile 2 (1.5–2.5 h)	3.35	3.26–3.44	< 0.01
ED LOS quartile 3 (2.5–4.4 h)	3.97	3.89–4.06	< 0.01
ED LOS quartile 4 (< 4.4 h)	4.97	4.89–5.05	< 0.01

Abbreviations: IHCA = in-hospital cardiac arrest; OR = odds ratio; CI = confidence interval; KTAS = Korean Triage and Acuity Scale; ED = emergency department; LOS = length of stay.

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
