# Peer review of "Prolonged Length of Stay in the Emergency Department and Increased Risk of In-Hospital Cardiac Arrest: A nationwide Population-Based Study in South Korea, 2016–2017"

_jcm, 2020, doi:10.3390/jcm9072284_

Round 1

Reviewer 1 Report

The authors sought to investigate whether prolonged emergency department (ED) length of stay (LOS) increased risk of in-hospital cardiac arrest in adults visiting a South Korean healthcare system. 

Major concerns:

Section 1. Introduction: the authors state "Because ED care focuses more on prompt diagnosis and stabilization than on detail care of the patient, as in the intensive care unit (ICU) or admission as an inpatient, a more extended stay in the ED could increase the risk of adverse events." Are you implying that patients with shorter LOS are prevented from having a cardiac arrest at all? If so, please explain the logic behind this statement. Is ED care not sufficient to prevent cardiac arrests compared to in-patient unit or ICU admissions?

Section 2. Materials and Methods, 2.2 Selection of participants: why were cardiac arrests within 60 minutes of ED admission excluded, when median EDLOS was only 2.5 hours, and most arrests occurred within 2 hours? Please explain the logic of these exclusions, or perform a subanalysis comparing early (60 minutes or less) to late (60+ minutes) cardiac arrests.

Author Response

Reviewer 1:

The authors sought to investigate whether prolonged emergency department (ED) length of stay (LOS) increased risk of in-hospital cardiac arrest in adults visiting a South Korean healthcare system. 

Major concerns:

Section 1. Introduction: the authors state "Because ED care focuses more on prompt diagnosis and stabilization than on detail care of the patient, as in the intensive care unit (ICU) or admission as an inpatient, a more extended stay in the ED could increase the risk of adverse events." Are you implying that patients with shorter LOS are prevented from having a cardiac arrest at all? If so, please explain the logic behind this statement. Is ED care not sufficient to prevent cardiac arrests compared to in-patient unit or ICU admissions?

Response> Thank you for valuable comments. We totally agreed with your suggestion that the statement you mentioned had not been supported by reasonable evidences. We did not mean that the patients with shorter LOS avoided a cardiac arrest at all. Meanwhile, caring critically ill patients is strive, and requires delicate monitoring and interventions. Patients those who prolonged stay in ED can be exposed in negative environment and tend to be resulted in adverse outcomes in numerous situations (Hong YC, et al. Am J Emerg Med 2009;27:385-390, Hung SC, et al. Crit Care 2014;1-9). Previous reports announced that longer ED LOS might result in increased in-hospital mortality (Chalfin DB, et al. Crit Care Med 2007;35:1477-1483, Mullin PM et al. Acad Emerg Med 2013;20:479-486, Ewart GW, et al. Chest 2004;125:1518-1521). Although the environments and situations are unlike according to each hospital, emergency physicians and nurses are rarely educated in care of the critically ill patients. Moreover, patient-to-nurse ratios are quite higher in the ED than in the ICU that may lead to poor medical services. We added theses contents in the manuscript as below.

“Caring critically ill patients is strive, and requires delicate monitoring and interventions. Because ED care focuses more on prompt diagnosis and stabilization than on detail care of the patient, as in the intensive care unit (ICU) or admission as an inpatient, a more extended stay in the ED could increase the risk of adverse events [5]. Moreover, emergency physicians and nurses are rarely educated in care of the critically ill patients, and patient to nurse ratio are quite higher in the ED than in the intensive care unit (ICU) that may lead to poor medical services [5].” (line 48-54)

Section 2. Materials and Methods, 2.2 Selection of participants: why were cardiac arrests within 60 minutes of ED admission excluded, when median EDLOS was only 2.5 hours, and most arrests occurred within 2 hours? Please explain the logic of these exclusions, or perform a subanalysis comparing early (60 minutes or less) to late (60+ minutes) cardiac arrests.

Response> We agreed with your concern that excluding patients who had cardiac arrest within 60 minutes was not clear. The primary purpose of our study evaluated the association between the EDLOS and the occurrence of unexpected IHCAs. We assumed that the development of cardiac arrest within one hour did not be influenced by prolonged LOS. Moreover, most cardiac arrests that occurred within 60 minutes were initially unstable and could not be avoidable, although physicians tried to manage and resuscitate patients. However, we also accepted that 1 hour was arbitrary and addressed this in the limitation section. Moreover, according to the reviewer’s suggestion, we performed an additional analysis comparing early to late cardiac arrests as your recommendations. Developments of cardiac arrest within one hour were 874 (8.7% of total IHCA), and comparisons between these patients and patients without IHCA within one hour showed a similar trend that that of CA after one hour (Table S1). Table S2 showed patients suffering from IHCA after one hour had a lower proportion of GCS < 13, KTAS level 1, and ICU admissions than that of IHCA within one hour. However, we also accepted that 1 hour was arbitrary and addressed this in the limitation section.

“Third, we excluded the patients (874, 8.7% of total IHCA) those who experienced IHCA within one hour because the primary purpose of our study was to reveal the association between prolonged LOS and adverse outcomes. We assumed that the development of IHCA within one hour did not be influenced by prolonged ED stay. Furthermore, most cardiac arrests occurred within 60 minutes were initially unstable and could not be avoidable although physicians tried to manage and resuscitate patients. Group of IHCA after 1 hour had a lower proportion of Glasgow Coma Scale < 13, KTAS level 1, and ICU admissions than that of IHCA within 1 hour, and these results supported our assumptions (Table S1, S2).” (line 238-245)

Table S1. Baseline demographic and clinical characteristics of emergency department patients with and without in-hospital cardiac arrest (IHCA) within one hour

Variables

Total

(N = 832,226)

No IHCA

(N = 831,353)

IHCA

(N = 873)

p-value

Age

45.0 (32.0 – 59.0)

45.0 (32.0 – 59.0)

63.0 (53.0 – 75.0)

< 0.01

Male

404,230 (48.6)

403,591 (48.5)

639 (73.2)

< 0.01

MAP (mmHg)

93.3 (86.7 – 103.7)

93.3 (86.7 – 103.7)

90.7 (73.0 – 106.7)

< 0.01

Pulse rate, bpm

80.0 (72.0 – 88.0)

80.0 (72.0 – 88.0)

92.0 (76.0 – 110.0)

< 0.01

Respiratory rate, breaths per min

20.0 (18.0 – 20.0)

20.0 (18.0 – 20.0)

20.0 (18.0 – 22.0)

< 0.01

LOS in ED (hours)

0.5 (0.2 – 0.8)

0.5 (0.2 – 0.8)

0.7 (0.6 – 0.9)

< 0.01

Cancer

23,071 (2.8)

23,035 (2.8)

36 (4.1)

< 0.01

Not alert††

25,995 (3.1)

25,458 (3.1)

537 (61.5)

< 0.01

KTAS level 1

18,816 (2.3)

18,274 (2.2)

542 (62.1)

< 0.01

ICU admission

17,927 (2.2)

17,289 (2.1)

638 (73.1)

< 0.01

Mortality

3,346 (0.4)

3,136 (0.4)

210 (24.1)

< 0.01

Data are presented as median (interquartile range [IQR]) or as number (percentage).

Vital signs were checked on ED arrival.

†† Glosgow Coma Scale < 13.

Abbreviations: IHCA = in-hospital cardiac arrest; MAP = mean arterial pressure; ED = emergency department; KTAS = Korean triage acuity scale; ICU = intensive care unit.

Table S2. Baseline demographic and clinical characteristics of emergency department patients with in-hospital cardiac arrest (IHCA) within and after one hour

Variables

After 1 hour

(N = 9,180)

Within 1 hour

(N = 873)

p-value

Age

68.0 (55.0–78.0)

63.0 (53.0 – 75.0)

< 0.01

Male

6,036 (65.8)

639 (73.2)

< 0.01

MAP (mmHg)

87.3 (70.0–104.0)

90.7 (73.0 – 106.7)

< 0.01

Pulse rate, bpm

95.0 (78.0–114.0)

92.0 (76.0 – 110.0)

< 0.01

Respiratory rate, breaths per min

20.0 (18.0–22.0)

20.0 (18.0 – 22.0)

< 0.01

LOS in ED (hours)

4.2 (2.4–9.3)

0.7 (0.6 – 0.9)

< 0.01

Cancer

1,032 (11.2)

36 (4.1)

< 0.01

Not alert††

5,006 (54.5)

537 (61.5)

< 0.01

KTAS level 1

4,416 (48.1)

542 (62.1)

< 0.01

ICU admission

5,375 (58.6)

638 (73.1)

< 0.01

Mortality

3,277 (35.7)

210 (24.1)

< 0.01

Data are presented as median (interquartile range [IQR]) or as number (percentage).

Vital signs were checked on ED arrival.

†† Glosgow Coma Scale < 13.

Abbreviations: IHCA = in-hospital cardiac arrest; MAP = mean arterial pressure; ED = emergency department; KTAS = Korean triage acuity scale; ICU = intensive care unit.

Reviewer 2 Report

Dear Authors,

you have presented interesting paper about cardiac arrest, but I have some minor and major comments and suggestions about it.

Abstract – as in instructions for Authors - The abstract should be a total of about 200 words maximum. The abstract should be a single paragraph and should follow the style of structured abstracts, but without headings.

line 51 – the reference is in the middle of sentence but should be at the end.

Please explain what are the codes I469W1 and I469W2, in what kind of ICD you have it?

There should be abbreviations of number, percentage and mean in section statistical analysis.

Was normality test performed, if yes add information about it?

In section Materials and Methods should be presented information about exclusion cases not in Results section.

You presented the limitation of your research, but you do not give no information about reason of what the patients were admitted to the emergency department.

References have to be modified due to Instructions for Authors of the Journal.

Author Response

Reviewer 2:

Dear Authors,

you have presented interesting paper about cardiac arrest, but I have some minor and major comments and suggestions about it.

Abstract – as in instructions for Authors - The abstract should be a total of about 200 words maximum. The abstract should be a single paragraph and should follow the style of structured abstracts, but without headings.

Response> As you commented, we revised the abstract following the guidelines.

Abstract: This study was to determine whether prolonged emergency department (ED) length of stay (LOS) is associated with increased risk of in-hospital cardiac arrest (IHCA). Retrospective cohort with a nationwide database of all adult patients visited to EDs in South Korea between January 2016 and December 2017 was performed. A total of 18,217,034 patients visited an ED during the study period. The median ED LOS was 2.5 hours. IHCA occurred in 9,180 patients (0.2%). IHCA was associated with longer ED LOS (4.2 vs. 2.5 hours), and higher rates of intensive care unit (ICU) admission (58.6% vs. 4.7%) and in-hospital mortality (35.7% vs. 1.5%). The ED LOS correlated positively with the development of IHCA (Spearman ρ = 0.91; p < 0.01) and was an independent risk factor for IHCA (odds ratio [OR] 1.10; 95% confidence interval [CI], 1.10–1.10). The development of IHCA increased in a stepwise fashion across increasing quartiles of ED LOS, with ORs for the second, third, and fourth relative to the first being 3.35 (95% CI, 3.26–3.44), 3.974 (95% CI, 3.89–4.06), and 4.97 (95% CI, 4.89–5.05), respectively. ED LOS should be reduced to prevent adverse events in patients visiting the ED.” (line 27-38)

line 51 – the reference is in the middle of sentence but should be at the end.

Response> As you mentioned, we deleted and rewrote the reference at the right position.

“The incidence of IHCA in adults has been reported to range from six to eight cases per 1,000 admissions, although the true incidence remains unclear[2,3].” (line 45-47)

Please explain what are the codes I469W1 and I469W2, in what kind of ICD you have it?

Response> Thank you for your valuable comments. We used ICD 10 codes for detecting cardiac arrests. I469W1 and I469W2 were codes only using in the study facility. We deleted these two codes for making clear and added the additional explanations about the codes I46.0 and I46.1 in the manuscript to clarify the meanings.

“Cardiac arrest was identified using ICD 10 codes corresponding to cardiac arrest or resuscitation, such as I46.0 (cardiac arrest with successful resuscitation), I46.1 (sudden cardiac death), and I46.9 (cardiac arrest, unspecified).” (line 80-82)

There should be abbreviations of number, percentage and mean in section statistical analysis.

Response> We corrected the sentences and inserted abbreviations.

“Continuous variables are presented as median with interquartile ranges (IQR), and compared by Student’s t-test or the Mann–Whitney U-test, as appropriate. Categorical variables are expressed as number (N) and percentage (%), and compared by chi-square tests.” (line 112-114)

Was normality test performed, if yes add information about it?

Response> We performed the normality test via using the Kolmogorov-Smirnov test with histogram. The all Z values were below 0.05, therefore, we performed non-parametric tests for the numeric and categorical variables.

“The normality of distribution was examined using the Kolmogorov-Smirnov test.” (line 111-112)

In section Materials and Methods should be presented information about exclusion cases not in Results section.

Response> Thank you for your correction. We inserted the exclusion criteria in the Materials and Methods section.

“In addition, we excluded patients who were under 18 years old; those with trauma; and those without any vital signs during ED stay.” (line 88-89)

You presented the limitation of your research, but you do not give no information about reason of what the patients were admitted to the emergency department.

Response> We agreed your opinion and reported ten common diagnosis of ED admitted patients like below.

“We categorized top 10 classification of ED diagnosis according to the development of IHCA and found that primary respiratory and cardiac causes were dominant (Table S3).” (line 236-238)

Table S3. Top 10 categories for ED diagnosis

No IHCA (N = 5,817,436)

IHCA (N = 9,180)

Diagnosis

Frequency

Diagnosis

Frequency

Gastrointestinal

890,185 (15.3)

Primary cardiac (non-AMI)

2,383 (26.0)

Nontraumatic neurologic disorder

471,379 (8.1)

Primary respiratory

1,629 (17.7)

Primary respiratory

219,586 (3.8)

AMI

1,377 (4.1)

Urology

171,588 (2.9)

Others

154 (1.7)

Unspecified fever

108,365 (1.9)

Sepsis

153 (1.7)

Others

94,885 (1.6)

Gastrointestinal

142 (1.3)

Urticaria

77,566 (1.3)

Cerebrovascular

134 (1.4)

Dyspnea

51,611 (0.9)

Chronic kidney disease

75 (0.1)

Primary cardiac

49,089 (0.1)

Trauma

61 (0.1)

Trauma

46,565 (0.1)

Acute Kidney Injury

58 (0.1)

Data are presented as number (percentage).

Abbreviations: ED = emergency department; IHCA = in-hospital cardiac arrest; AMI = acute myocardial infarction.

References have to be modified due to Instructions for Authors of the Journal.

Response> Thank you. We checked references again following the instructions of the journal.